# Echocardiographic Evaluation of the Cardiac Chambers in Asthmatic Patients: The BADA (Blood Pressure Levels, Clinical Features and Markers of Subclinical Cardiovascular Damage of Asthma Patients) Study-ECO

**DOI:** 10.3390/jpm12111847

**Published:** 2022-11-05

**Authors:** Domenico Di Raimondo, Gaia Musiari, Giuliana Rizzo, Edoardo Pirera, Alida Benfante, Salvatore Battaglia, Daniela Colomba, Antonino Tuttolomondo, Nicola Scichilone, Antonio Pinto

**Affiliations:** 1Division of Internal Medicine and Stroke Care, Department of Health Promotion, Mother and Child Care, Internal Medicine and Medical Specialties (ProMISE) “G. D’Alessandro”, University of Palermo, Piazza delle Cliniche, 90127 Palermo, Italy; 2Division of Respiratory Diseases, Department of Health Promotion, Mother and Child Care, Internal Medicine and Medical Specialties (ProMISE) “G. D’Alessandro”, University of Palermo, Piazza delle Cliniche, 90127 Palermo, Italy

**Keywords:** asthma, cardiovascular risk, echocardiography, cardiac damage, essential hypertension, inhaled corticosteroid therapy, beta-adrenergic agonist agents

## Abstract

The “Blood pressure levels, clinical features and markers of subclinical cardiovascular Damage of Asthma patients” (BADA) study is aimed at defining the cardiovascular risk profile and the markers of subclinical and clinical vascular and cardiac damage in asthmatic patients. Very few studies have assessed asthmatic patients without concomitant heart disease through a transthoracic echocardiogram. The goal of the present study is to investigate the prevalence of morphology and/or function changes in the cardiac chambers of a sample of 86 patients with chronic asthma, referred to the dedicated outpatient unit of the Division of Respiratory Diseases of the AOUP “P. Giaccone” of the University of Palermo, and the results obtained were compared with those of a control group without respiratory or cardiovascular diseases. Patients with asthma showed a marked and widespread involvement of the four cardiac chambers compared with the controls: enlargement of the two atria, greater left ventricular remodeling with interventricular septal thickening, increased indexed left ventricular mass with a significantly greater percentage of patients with overt left ventricular hypertrophy, worse left ventricular diastolic function proven by the significant difference in the E/A ratio, and worse right ventricular systolic function with global right ventricular dysfunction estimated by the Myocardial Performance Index (Tei Index). Multivariate regression analysis, after adjustment for essential hypertension, hypertension severity, diabetes, Body Mass Index, and creatinine clearance, seems to indicate that the indexed left ventricular mass, right atrial volume, and right ventricular Tei index (but not left ventricular hypertrophy) correlate significantly with asthma, severe asthma, and FEV_1_ (and to a lesser extent with asthma duration). No correlation is apparent between inhaled therapy (ICS, SABA) and myocardial involvement. These results seem to confirm that a more in-depth cardiovascular evaluation in patients with chronic respiratory disease allows the identification of unrecognized cardiovascular involvement. A transthoracic echocardiogram performed in asthmatic patients without clinically overt signs or symptoms of cardiovascular impairment has identified some features indicative of an early subclinical cardiac impairment not found in the control group. These findings, considering also the higher frequency of hypertension in the asthma group, deserve further validation in the future.

## 1. Introduction

BADA (blood pressure levels, clinical features, and markers of subclinical Cardiovascular damage of asthma patients) is a study that started a few years ago with the main objective of better defining the 24 h blood pressure profile, cardiovascular risk, and markers of subclinical and clinical vascular and cardiac damage in asthmatic patients. Our previous results [1] suggested the wide use of 24 h ambulatory blood pressure monitoring in subjects with asthma, reporting that arterial hypertension in this population is more prevalent than what has been assumed. Very few studies have assessed, through a transthoracic echocardiogram (TTE), asthmatic patients without concomitant heart disease. Tuleta et al. [2] in 2018 recruited 72 subjects with asthma and different degrees of disease severity, encompassing both mild–moderate and severe forms, comparing them with 20 healthy controls. The sample was also divided into different subgroups to assess the possible role of chronically taken drug therapy in cardiac remodeling. Speckle tracking analysis, based on longitudinal strain reduction, revealed an increased incidence of subclinical biventricular heart failure. Sobhy et al. [3], in a small study that enrolled 30 asthmatic adults, demonstrated an early systolic and diastolic dysfunction of the right ventricle. In the Bogalusa Heart Study [4], a prospective study performed among 1118 young adults (average age at follow-up 36.7 ± 5.1 years), after a mean follow-up of 10.4 ± 7.5 years, a greater increase in the left ventricle (LV) mass (167.6 vs. 159.9 g; *p* = 0.01) as well as in the left ventricular mass indexed for body surface (LVMi) (40.7 vs. 37.7 g/m^2^; *p* < 0.01) was observed in asthmatic patients than in the control group of non-asthmatic subjects, regardless of age, gender, ethnicity, active or non-active smoking status, systolic blood pressure, or whether antihypertensive medication was used.

In view of the scarcity of available data, the aim of the current study is to ascertain the prevalence of alterations in the morphology and/or function of the cardiac chambers in a sample of asthmatic patients, comparing the results obtained with those of a control group without respiratory or cardiovascular diseases.

## 2. Materials and Methods

The data reported in this study were collected on 86 patients with chronic asthma, referred to the dedicated outpatient unit of the Division of Respiratory Diseases of the AOUP “P. Giaccone” of the University of Palermo (Italy) in the period between 1 January 2018 and 1 June 2020. All subjects enrolled were Caucasian, were previously informed about the characteristics and the object of the study, and a written informed consent was obtained. The BADA study was approved by the local ethical committee (ref n. 018/2017).

### 2.1. Exclusion Criteria

-Coexistence of Chronic Obstructive Pulmonary Disease (COPD);-Coexistence of Obstructive Sleep Apnea Syndrome (OSAS);-Active smoking status;-History of chronic ischemic heart disease, valve disease, or other structural heart disease with or without secondary impairment of left ventricular ejection fraction;-History, in the six months preceding the study, of any acute myocardial, endocardial, or pericardial pathology capable of transiently or permanently compromising cardiac structure and/or function.

A control group of 100 subjects was created; patients were enrolled among those referring to the Division of Internal Medicine with Stroke Care of the University of Palermo for an outpatient visit due to different reasons, not affected by acute or chronic respiratory or cardiovascular disease. The subjects enrolled in this group, in particular, denied having a clinical history of bronchial asthma or active use of steroid therapy for any reason and did not presented spirometry evidence of respiratory flow limitations.

The initial study procedure included, for all subjects, a comprehensive recording of medical history with specific attention to the cardiovascular risk profile, pharmacological anamnesis, complete physical examination, assessment of Body Mass Index (BMI), calculated as the individual’s body weight divided by the square of his or her height, and the following blood biochemical examinations: total cholesterol, High Density Lipoprotein (HDL) cholesterol, triglyceride, creatinine, fibrinogen, complete blood count, fasting glucose, and C-reactive protein. Subjects were defined as type 2 diabetics, if they had known diabetes treated by diet, oral hypoglycemic drugs, or insulin. Previous cerebrovascular disease: Transient Ischemic Attack (TIA) or ischemic stroke was assessed by history, specific neurological examination performed by specialists, and hospital or radiological (brain computed tomography or brain magnetic resonance) records of definite previous stroke. Previous coronary heart disease was detected by history, clinical examination, electrocardiogram, and echocardiogram. All subjects enrolled were self-reported as being sedentary.

### 2.2. Respiratory Evaluation

The enrolled asthmatic patients met the diagnosis of the disease based on GINA criteria [5], considering the history of respiratory symptoms and the functional evidence of reversible bronchial obstruction based on the spirometry examination. Lung function tests were performed according to ERS/ATS recommendations [6]. The analysis included the following parameters: FEV_1_ (forced expiratory volume in 1 s) expressed in liters and in percent predicted value (FEV_1_%), FVC (forced vital capacity) expressed in liters and in percent predicted value (FVC%), and the FEV_1_/FVC ratio, expressed as absolute values. Reversibility of bronchial obstruction was assessed.

Severe asthma was defined as asthma that requires treatment with high-dose inhaled corticosteroids (ICS) plus a second controller (and/or systemic corticosteroids) or that remains uncontrolled despite this therapy [5]. The severity of the disease was assessed according to the treatment level to control asthma symptoms at the time of the first examination.

The quantification of the daily intake of ICS, in relation to patient age (in mcg/day), was obtained according to the clinical comparability table indicated by the GINA document [5].

### 2.3. Echocardiographic Evaluation

A transthoracic echocardiogram was performed in all enrolled patients. For all the subjects enrolled, the same GE-Vivid 7 echocardiographic device was used by the same experienced echocardiographer. The patient was placed in the supine position and in left lateral decubitus; data were stored digitally for offline evaluation using a specific program, to be subsequently interpreted by the same echocardiographer who performed the exam. All measurements and echocardiographic data were collected and analyzed in accordance with the latest guidelines of the American Society of Echocardiography [7].

The echocardiographic parameters evaluated in the present analysis, summarized in Figure 1, were the following:

### 2.4. For the Left Ventricle (LV)

-Left ventricular end-diastolic volume measured in ml (LV-EDV). We considered reference values ranging from 42 and 58.4 mL for men and 37.8 and 52.2 mL for women.-Interventricular septum thickness (IVS). We considered reference values ranging from 6 and 10 mm for men and 6 and 9 mm for women.-Relative wall thickness of the left ventricle (LV-RWT). RWT was calculated by the formula [(2 × PWTd)/(LVIDd)], where PWTd is the posterior wall thickness of the LV and LVIDd is the telediastolic inner diameter of the LV.-Left atrium volume indexed for body surface (LAVI). The volume obtained by measuring atrial areas and diameters was indexed for body surface area, and a left atrial volume of up to 34 mL/m^2^ was considered normal in both genders.-Left ventricular mass indexed for body surface (LVMI). We considered a normal LVMI in the case of values between 49 and 115 g/m^2^ for men and between 43 and 95 g/m^2^ for women.-Left ventricular hypertrophy (LVH). We defined a subject having LVH when LVMI values exceed 115 g/m^2^ in men and 95 g/m^2^ in women.-Left ventricular ejection fraction (LVEF). We considered normal a LVEF > of 52% for men and >54% for women.-E/A ratio: Pulsed Doppler images at the mitral cusp level were used to measure transmitral flow velocities; these were measured (in m/s) during the passive early diastolic filling peak (early, E-wave) and during the late diastolic flow peak provided by atrial contraction (A-wave). The E/A ratio was then calculated, which describes the blood flow from the atria to the ventricles during ventricular diastole and provides information on the atrial contribution to ventricular filling. The value of the E/A ratio in a subject with normal diastolic function is between 0.8 and 2. Three abnormal patterns were defined:-Impaired relaxation pattern or grade 1 diastolic dysfunction is the mildest degree of diastolic dysfunction (E/A < 1).-Pseudonormal mitral inflow pattern or grade 2 diastolic dysfunction has a falsely normal E/A ratio (between 0.8 and 1.5).-Restrictive pattern or grade 3 diastolic dysfunction, in which an E/A ratio >2 indicates a particularly severe impairment of ventricular compliance.-E/e’ Ratio. A Pulsed Tissue Doppler of the mitral annulus was used to measure the early protodiastolic peak (e’) velocities of the septal and lateral mitral annulus as well as the velocities of the lateral tricuspidal annulus; all velocities are expressed in cm/sec. The E/e’ ratio was then calculated using the mean value of e’ of the septal versant lateral side of the mitral valve. In accordance with the American Society of Echocardiography Recommendations for the Evaluation of LV Diastolic Function [8,9], the final assessment of diastolic dysfunction considered mitral E, E/e’ ratio, and E/A ratio. An E/e’ ratio less than 8 usually indicates normal filling pressures, whereas a ratio >15 suggests increased filling pressures (ventricular diastolic dysfunction) even in the case of an E/A ratio >1. A value between 8 and 15 needs further investigation to be correctly interpreted.

### 2.5. For the Right Ventricle (RV)

-Right atrium volume indexed for body surface (RAVi). The size of the RA was assessed using the apical 4-chamber projection. From this window, the area of the right atrium was measured by planimetry. The area of the right atrium was plotted in ventricular systole (when atrial volume is maximal) excluding the area between the tricuspid flaps and the annulus, following the atrial endocardium. Volume was calculated from the data collected, and a right atrial volume of 18 to 32 mL/m^2^ in men and 15 to 27 mL/m^2^ in women was considered normal.-Tricuspid annular plane systolic excursion (TAPSE). It is measured in M-Mode using the apical 4-chamber projection. TAPSE quantifies the systolic excursion of the tricuspid annulus along the longitudinal plane, thus representing an index of the efficiency of right ventricular systolic function. The greater the excursion, the better the performance of the right ventricle. Although this index essentially describes RV longitudinal function, it has shown a good correlation with other parameters that estimate global RV systolic function, such as myocardial scintigraphy and 2D estimation of RV ejection fraction. A value >17 mm was considered normal in both sexes.-Right ventricular ejection fraction (RVEF). As with LVEF, it was calculated as a percentage of the volume difference. Although it is most appropriately conducted, it is a parameter that suffers from great variability in relation to possible acute/chronic RV overload, paradoxical left/right septal motion, or poor acoustic window. A right ventricular systolic function >45% was considered normal in both sexes.-Myocardial Performance Index (Tei Index). It was calculated for both ventricles by tissue Doppler, using the formula IVCT + IVRT/ET, where IVCT is the isovolumetric contraction time, IVRT is the isovolumetric release time, and ET is the ejection time. It is an index that incorporates both systolic and diastolic time intervals and is therefore capable of globally expressing both diastolic and systolic function. Systolic dysfunction prolongs projection (the isovolumetric contraction time, ICVT) and reduces ejection time (ET). Both systolic and diastolic dysfunction result in abnormal myocardial release, prolonging the release period (isovolumetric release time, IVRT). A value >0.55 was considered normal for the RV, while a value of 0.39 ± 0.05 was considered normal for the LV [8,9].

### 2.6. Statistical Analysis

The statistical analysis of quantitative and qualitative data, including descriptive statistics, was performed for all items. Continuous data are expressed as mean ± 1 standard deviation; unless otherwise specified, categorical variables are expressed as percentage. A Student’s *t*-test was used to compare the continuous variables between the two groups examined. The comparison of the proportions was performed using the chi-squared test or the exact Fischer’s test (used when the expected frequency of the event was lower than five times).

A multivariate regression analysis was performed to evaluate the presence and strength of correlation between asthma and some of the main factors related to it (severe asthma, duration of asthma, FEV_1_%, average daily dosage of inhaled corticosteroids, and average daily dosage of short-acting beta-agonists) with the main four echocardiographic alterations that we showed to be associated with asthma and that we judged to be emblematic for the evaluation of the association between asthma and myocardial involvement (indexed left ventricular mass, presence of left ventricular hypertrophy, and right atrial volume, right ventricular Tei index). All the results of the multivariate analysis were adjusted for the following confounding factors: essential hypertension, diabetes, Body Mass Index, and creatinine clearance.

All *p*-values were two-sided and *p*-values less than 0.05 were considered statistically significant. Statistical analysis and graphs generation were performed using R software, a free software developed by The R Foundation (https://www.r-project.org/), version 4.1.2.

## 3. Results

Table 1 shows the main anthropometric, clinical, and laboratory variables analyzed in the 86 asthmatic subjects and in the 100 control subjects without respiratory and cardiovascular diseases, compared by sex and age. With a comparable family history of hypertension, the percentage of asthmatics with hypertension was significantly higher than that of the controls: 57.5% (50 subjects) vs. 35.0% (35 subjects) (*p*: 0.01) as well as mean Systolic Blood Pressure levels (132.9 ± 22,6 vs. 124.9 ± 15.3 mmHg, *p*: 0.05). The two groups did not differ in relation to the main comorbidities/risk factors associated with hypertension, except for a higher prevalence of type 2 diabetes mellitus (DM) in the asthma group (8 patients vs. 0) (*p*: 0.001), despite a concordance of fasting blood glucose values in asthmatics vs. controls (90. 9 vs. 86.9 mg/dL) (*p*: 0.07). In asthmatics, we found a worse renal function (estimated as creatinine clearance calculated according to Cockcroft’s and Gault’s method) (66.83 vs. 88.12 mL/min) (*p*: 0.01). Body Mass Index was higher in cases than in the controls (28.11 vs. 24.22) (*p*: 0.05). Spirometry indexes of asthmatic patients were within the normal range, although FEV_1_ and FEV_1_/FVC ratio were mildly but significantly reduced compared with the healthy controls (respectively 79.4 ± 13.3 vs. 93.2 ± 5.1, *p*: 0.01 and 84.3 ± 11.4 vs. 95.5 ± 19.2, *p*: 0.05).

Table 2 shows the analysis of the indexes obtained from the echocardiographic evaluation in asthmatic patients and the controls.

Our results show that asthma affects the structure and function of all four cardiac chambers. We observed statistically significant differences for: LV-EDV: 50.14 ± 3.38 vs. 66.51 ± 2.66 mL (*p*: 0.01); IVS thickness: 11.33 ± 1.55 vs. 7.57 ± 0.87 mm (*p*: 0.01); LVMi: 99.48 ± 33.11 vs. 89.22 ± 26.19 g/m^2^ (*p*: 0.01); LV RWT: 0.37 ± 0.21 vs. 0.31 ± 0.12 (*p*: 0.045); LAVi: 40.23 ± 4.47 vs. 32.34 ± 5.51 mL/m^2^ (*p*: 0.001); the percentage of subjects who exceeded the threshold for the definition of LVH (48.8% of asthmatics vs. 12% of controls; *p* < 0.0001); LV diastolic function estimated by the mitral E/A ratio (1.03 ± 0.21 vs. 1.46 ± 0.17; *p*: 0.03); RAVi: (32.69 ± 7.07 vs. 26.41 ± 4.55 mL/m^2^; *p*: 0.01); RV systolic function measured by the ejection fraction (42.9% vs. 50.5%; *p*: 0.05); and global right ventricular function assessed by the Tei index (0.56 vs. 0.62; *p*: 0.04).

Table 3 show the results of the multivariate regression analysis performed for the following traits: severe asthma, duration of asthma, FEV1%, average daily dosage of inhaled corticosteroids, average daily dosage of short-acting beta-agonists, and four of the echocardiographic alterations that were found in our analysis to be associated with asthma condition and that we considered representative for the independent association between asthma and myocardial involvement (indexed left ventricular mass, left ventricular hypertrophy, right atrial volume, and right ventricular Tei index). Asthma, severe asthma, and FEV1% levels were independently correlated with all echocardiographic parameters examined, except for left ventricular hypertrophy. All results of multivariate analysis are adjusted for essential hypertension, hypertension severity (estimated through the level of “uncontrolled hypertension”, i.e., the failure to achieve the targets recommended by the 2018 ESC-ESH guidelines), diabetes, Body Mass Index, and creatinine clearance.

## 4. Discussion

The main findings of our study, one of the few to have performed a transthoracic echocardiographic evaluation in a cohort of asthmatic subjects without respiratory and cardiovascular comorbidities, comparing them with a control group comparable in sex and age, are the following. Patients with asthma have marked and widespread involvement of the four cardiac chambers compared with the controls: enlargement of the two atria, greater left ventricular remodeling with interventricular septal thickening, increased indexed left ventricular mass, and a significantly greater percentage of patients with overt left ventricular hypertrophy; worse diastolic function shown by the significant difference in the E/A ratio; and right ventricular systolic dysfunction with less valid global right ventricular function estimated by the Tei index. Multivariate regression analysis, after adjustment essential hypertension, hypertension severity, diabetes, Body Mass Index, and creatinine clearance, confirmed that asthma, severe asthma, and FEV_1_% (and to a lesser extent asthma duration) correlated significantly with indexed left ventricular mass, right atrial volume, and right ventricular Tei index (but not with the presence of left ventricular hypertrophy). No correlation was instead shown between inhalation therapy (ICS, SABA) and myocardial involvement.

Our data on LV structure and function in asthmatics highlight a significant increase in LV thickness, LV mass, and a four-fold prevalence of LV hypertrophy compared with the non-asthmatic controls. In our study, increased LV mass appears to be one of the pivotal elements of the cardiac remodeling in asthmatic patients. LV remodeling and thickening is commonly the result of chronic volume and/or pressure overload, with an estimated prevalence in the general population of 14.9% in men and 9.1% in women, respectively, and an acme of 36 to 41% in subjects with high blood pressure [10]. LVMi is an independent risk factor for progression to reduced ejection fraction heart failure (HFrEF) but also for the development of death from cardiovascular events [11]. Other studies, such as the Bogalusa Heart Study [4], have strengthened, before ours, the hypothesis underlying the relationship between asthma, especially of long duration, and increased LV mass. Our sample of asthmatic subjects also showed a higher burden of comorbidities than the controls, which can further play a role in the progression of cardiovascular damage: a higher prevalence of hypertension (57.5% vs. 35.0% *p*: 0.013), diabetes mellitus (9.1% vs. 0% *p*: 0.005), higher BMI (27.3 vs. 25.8, *p*: 0.05), and lower creatinine clearance (66.8 vs. 97.6 mL/min *p*: 0.04). The special association existing between chronic respiratory disease (and asthma especially) with hypertension [1,12] particularly represents an additional factor that can condition the evolution of the asthmatic’s heart toward chronic pressure overload and compensatory increase in LV mass, although our data show that the association between asthma and increased LV mass is independent of arterial hypertension. Corroborating this, Deveraux et al. in the STRONG HEART Study [13] confirmed that the correlation between increased blood pressure, both for SBP and DBP, and increased LV mass was weaker than expected, concluding that more than half of the cases of myocardial alteration were related to nonhemodynamic factors. Our data, according to others [14], allow us to hypothesize that, in asthmatic subjects, the extent of lung impairment, evaluated through reduced levels of FEV_1_, FVC, and FEV_1_/FVC ratio, may contribute more than other determinants to the development of mild cardiac dysfunction. Despite this, it should be pointed out that hypertension is a disease more closely associated with asthma than suspected to date [1]; as a consequence, the cardiac involvement in asthmatics deserves more careful reevaluation in the future in this regard as well.

Higher BMI is associated with the development of LV hypertrophy [15] as well and being more prevalent in asthmatics than in the controls [1,5]. In our sample of subjects with asthma, the average BMI value (28.11 ± 4.09) falls within the overweight range, making it difficult to identify how significant the role also played by this co-factor is in the development of the cardiac involvement shown in our sample, also given the absence of comparable data. The analyses performed on our dataset suggest that, in asthmatics, hypertension, diabetes mellitus, or elevated BMI could act as co-factors in determining cardiac damage rather than as a primary cause, as might be assumed. These findings could further underscore the need for a closer clinical and instrumental monitoring of asthmatic patients in order to assess the cardiovascular risk and detect early onset of the progression of LV disfunction to HFrEF.

Our finding of a worse systolic and diastolic performance of the right ventricle in asthmatic subjects, although novel, has rather solid pathophysiological assumptions. Pulmonary artery wall tension, or pulmonary arterial stiffness (PAS), is a relatively recent noninvasive echocardiographic index aimed at assessing the structural and functional characteristics of the pulmonary vascular bed [16]. Primary bronchopulmonary diseases, such as bronchial asthma, through recurrent episodes of hypoxia, could result in increased PAS, through the remodeling of the pulmonary vascular structure. Particularly, an endothelial dysfunction caused by inflammation could be established from the early stages of the disease, which would in turn trigger the proliferation of smooth muscle cells (SMCs) as well as the deposition of matrix proteins in the context of the vessel wall, causing the degeneration of the elastic component of the pulmonary artery [17]. The increase in pulmonary stiffness would especially contribute to the genesis of ventricular dysfunction through an increase in the RV afterload, thus sharing the pathophysiology of various respiratory system disorders, which would in turn result in an increase in this index compared to the healthy control population. Only a few studies have evaluated the impairment of the right chambers in asthmatics [18,19], and only one study in adults [3], to the best of our knowledge; therefore; we have little data with which to compare our results.

The possible role of inhaled therapy taken daily by patients with asthma for many years in the genesis of cardiovascular complications is much debated. Several authors [1,20] have addressed the issue of the possible detrimental cardiovascular effects of chronic use of inhaled corticosteroids with inconclusive results. Sengstock et al. evaluated the role of chronic beta-agonist therapy on the future development of left ventricular dysfunction in asthmatics without showing a clear association [21]. Wang et al., in a large case–control study of more than 280,000 patients with COPD [22], concluded that the overall use of inhaled long-acting bronchodilators was not associated with an increased risk of CVD across different recency of therapy. On the other hand, both acute and chronic administration of beta2-agonist drugs in asthmatics appears to cause a shift in the sympathovagal balance toward sympathetic hypertone. This shift is associated with increased cardiac mortality and morbidity and increased risk of sudden coronary death [23]. Instead, Hirono et al. showed that the chronic use of beta-agonists in asthmatics is associated with a deterioration of predominantly diastolic left ventricular function. This dysfunction appears to be partially reversible after the discontinuation of the drug [24]. Our study seems to indicate that both ICS and SABA inhaled therapy does not affect the potential myocardial involvement. The multivariate regression analysis found no association between LABA or ICS and any of the indicators of cardiac alteration examined (left ventricular mass, left ventricular hypertrophy, right atrial volume, and right ventricular Tei index). However, interactions between chronic inhaled therapy and the cardiovascular system can occur in many ways, so caution in the use of these drugs is mandatory, especially avoiding the overuse of SABA [25].

A strength of our study is the choice of the sample: asthmatic adults without concomitant cardiovascular or respiratory disease. Many of the studies that have assessed, through echocardiography, the impact of asthma on cardiac function are conducted in children, adolescents, or teenagers [18,19]. Although these studies have shown diastolic dysfunction, likely the development of cardiac overload remodeling cannot be seen given the young age of the subjects enrolled. Another strength is the statistical analysis performed, which aimed to identify the role played by asthmatic disease and its associated features with possible cardiac involvement, independently of other confounders. Our study also has some limitations. First of all, the limited sample and the specific geographic area where the study was carried out does not allow the generalization of the results. Moreover, almost three-quarters of our sample can be categorized as having severe asthma: future studies should confirm the extent to which cardiovascular involvement is affected by asthma severity and disease duration as suggested by our data. Finally, in our study, patients in the asthma group differed from the controls in the higher prevalence of some cardiovascular risk factors (hypertension, diabetes, and high BMI), although they had overlapping age and lipid profile, and no one in the two groups was a smoker. Our statistical analysis seems to indicate an overriding role of asthma and spirometry indexes in the correlation with the main indicators of myocardial impairment. We cannot rule out the possibility that other factors related to the overall cardiovascular risk of the patients and not directly addressed in our analysis may have influenced the results, although our investigation was quite thorough.

In conclusion, in this second phase of the BADA study, we confirm that a more in-depth cardiovascular evaluation in patients with chronic respiratory disease, particularly in asthmatic patients, allows the identification of a significant subclinical level of cardiac involvement. A transthoracic echocardiogram performed in asthmatic patients without clinically overt signs or symptoms of cardiovascular impairment identified some features indicative of an early subclinical cardiac impairment not found in the control group. These findings, considering also the higher frequency of hypertension in the asthma group, deserves further validation in the future.

Our results, along with others currently being evaluated in the upcoming phases of the BADA study, could contribute to a more comprehensive framing of asthmatic patients, allowing the clinical-therapeutic approach to be optimized as well.

## Figures and Tables

**Figure 1 jpm-12-01847-f001:**
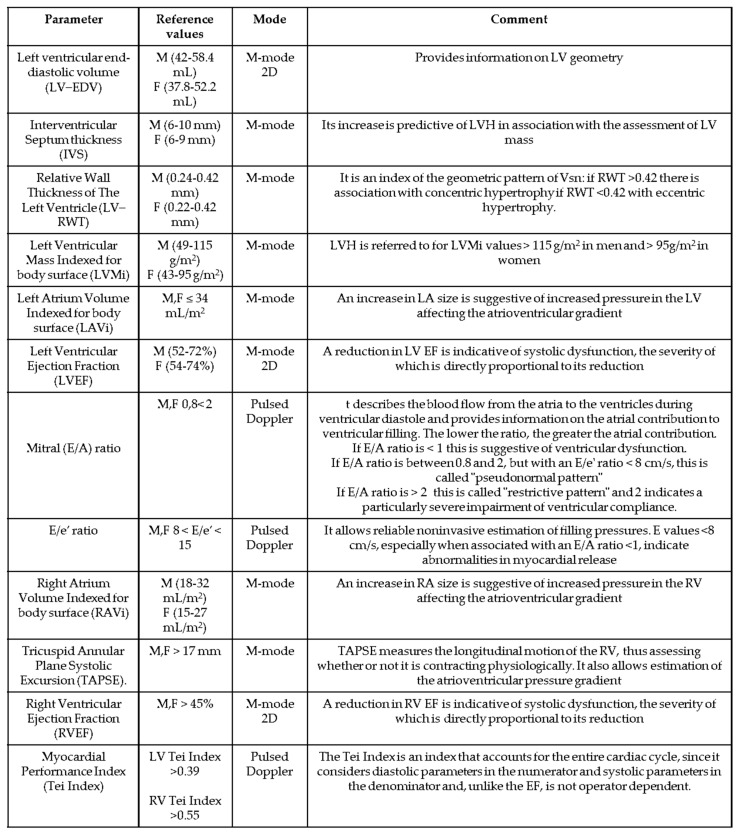
List and characteristics of the echocardiographic parameters evaluated in the study.

**Table 1 jpm-12-01847-t001:** Anthropometric, clinical, and laboratory variables: asthmatics vs. controls.

Variable	Asthmatics (*n*:86)	Controls (*n*:100)	*p*
M/F, *n* (%)	40/46 (46.5/53.5)	46/54 (46.0/54.0)	0.89
Age (y)	57.29 ± 14.81	56.47 ± 13.80	0.65
Hypertension, *n* (%)	50 (57.5)	35 (35.0)	0.01
Family history of hypertension, *n* (%)	60 (69.7)	62 (62.0)	0.11
Diabetes, *n* (%)	8 (9.1)	0 (0)	0.001
SBP	132.9 ± 22.6	124.9 ± 15.3	0.05
DBP	79.2 ± 12.8	75.8 ± 12.0	0.08
HR	77.1 ± 7.7	79.1 ± 8.1	0.25
Fasting glucose (mg/dl)	90.90 ± 9.75	86.95 ± 13.74	0.07
Past cerebral vascular event, *n* (%)	0 (0)	0 (0)	---
Past cardiac vascular event, *n* (%)	10 (11.6)	3 (3.0)	0.09
Past peripheral arterial disease, *n* (%)	0 (0)	0 (0)	---
Creatinine (mg/dl)	0.87 ± 0.36	0.71 ± 0.64	0.07
Creatinine clearance (mL/min)	66.83 ± 37.07	88.12 ± 19.24	0.01
Statin use, *n* (%)	13 (15.1)	5 (5.0)	0.06
BMI (Kg/m^2^)	28.11 ± 4.09	24.22 ± 5.06	0.05
Total cholesterol (mg/dl)	189.81 ± 38.08	206.22 ± 34.31	0.26
RBC (mm^3^)	4,630,000 ± 728,000	4,110,000 ± 880,000	0.09
Hb (g/dL)	12.7 ± 2.2	12.2 ± 2.6	0.11
MCV (fL)	87.9 ± 5.4	86.3 ± 4.4	0.62
WBC (mm^3^)	7348.7 ± 2939.5	8055.9 ± 3996.9	0.28
HDL cholesterol (mg/dl)	49.92 ± 17,68	47.14 ± 22.02	0.53
Triglycerides (mg/dl)	100.94 ± 38.20	96.88 ± 49.63	0.25
Current smokers, *n* (%)	0 (0)	0 (0)	---
Past smokers, *n* (%)	26 (30,2)	29 (29.0)	0.51
Severe asthma, *n* (%)	64 (74.8)	---	
Asthma duration (yrs)	15.49 ± 15.76	---	
Oral steroid therapy, *n* (%)	18 (20.9)	---	
ICS low dose, *n* (%)	0 (0)	---	
ICS medium dose, *n* (%)	5 (5.8)	---	
ICS/LABA low dose ICS (*n*/%)	25 (29.1)	---	
ICS/LABA medium dose ICS (*n*/%)	45 (52.3)	---	
ICS/LABA high dose ICS (*n*/%)	11 (12.8)	---	
Mean ICS daily dose (mcg)	560 ± 330	---	
Mean LABA daily dose (mcg)	32 ± 10	---	
LAMA (*n*/%)	33 (39.4)	---	
SABA (*n*/%)	39 (45.1)	---	
LTRA (*n*/%)	23 (27.5)	---	
Doxofylline (*n*/%)	14 (16.3)	---	
Biologic therapy (*n*/%)	5 (12.5)	---	
FEV_1_ (% predicted)	79.4 ± 13.3	93.2 ± 5.1	0.01
FVC (% predicted)	93.3 ± 5.9	97.2 ± 14.1	0.06
FEV_1_/FVC ratio	84.3 ± 11.4	95.5 ± 19.2	0.05
STEP GINA 2020 (*n*/%)		---	
1, 2	26 (30.2)		
3, 4	47 (54.6)		
5	13 (15.2)		

Data are presented as mean value ± SD; SBP: Systolic blood pressure; DBP: Diastolic blood pressure; HR: Heart rate; BMI: Body mass index; RBC: Red blood cells; Hb: Hemoglobin; MCV: Mean corpuscular volume; WBC: White blood cells; ICS: Inhaled corticosteroid; LABA: Long-acting beta-agonist; LAMA: Long-acting muscarinic antagonist; SABA: Short-acting beta-agonist; LTRA: Leukotriene receptor antagonist; FEV_1_: Forced expiratory volume in the 1st second; FVC: Forced vital capacity. Creatinine clearance was calculated through Cockcroft’s and Gault’s formula.

**Table 2 jpm-12-01847-t002:** Echocardiographic data of asthmatic vs. control subjects.

Variable	Asthmatics (*n*: 86)	Controls (*n*: 100)	*p*
LV-EDV (mL)	50.14 ± 3.38	66.51 ± 2.66	0.01
IVS thickness (mm)	11.33 ± 1.55	7.57 ± 0.87	0.01
LVMi (g/m^2^)	99.48 ± 33.11	89.22 ± 26.19	0.01
LV RWT	0.37 ± 0.21	0.31 ± 0.12	0.045
LAVi (mL/m^2^)	40.23 ± 4.47	32.34 ± 5.51	0.001
LVH (*n*/%)	42 (48.8)	12 (12.0)	<0.001
LV EF (%)	63.87 ± 4.50	66.25 ±5.15	0.74
E/A ratio	1.03 ± 0.21	1.46 ± 0.17	0.03
E/e’ ratio	10.06 ± 3.98	9.33 ± 4.66	0.14
LV Tei Index	0.52 ± 0.22	0.55 ± 0.14	0.67
RAVi (mL/m^2^)	13.1 ± 1.8	15.2 ± 2.8	0.01
TAPSE (mm)	19.78 ± 4.23	24.33 ± 3.54	0.06
RV EF (%)	42.9 ± 7.96	50.5 ± 5.66	0.05
RV Tei Index	0.56 ± 0.15	0.62 ± 0.12	0.04

Data are presented as mean value ± SD; LV-EDV: Left ventricular end-diastolic volume; IVS: Interventricular septum; LVMi: Left ventricular mass indexed; LV-RWT: Left ventricular relative wall thickness; LAVi: Left atrium volume indexed; LVH: Left ventricular hypertrophy; EF: Ejection fraction; RAVi: Right atrium volume indexed; TAPSE: Tricuspidal annular plane systolic excursion.

**Table 3 jpm-12-01847-t003:** Multivariate regression analysis.

Variable	β	R^2^	*p*
Correlation of asthma and left ventricular mass indexed			
Asthma ^1^	0.204	0.69	0.038
Severe Asthma ^1^	0.295	0.80	0.018
Asthma Duration ^1^	0090	0.14	0.173
ICS daily dose ^1^	0.115	0.21	0.107
SABA daily dose ^1^	0.055	0.10	0.233
FEV_1_% ^1^	−0.228	0.75	0.044
Correlation of asthma and right atrial volume			
Asthma ^1^	0.185	0.68	0.028
Severe Asthma ^1^	0.294	0.52	0.040
Asthma Duration ^1^	0.256	0.64	0.026
ICS daily dose ^1^	−0.114	0.19	0.218
SABA daily dose ^1^	0.044	0.10	0.440
FEV_1_% ^1^	−0.244	0.87	0.011
Correlation of asthma and right ventricle Tei Index			
Asthma ^1^	0.297	0.75	0.021
Severe Asthma ^1^	0.301	0.80	0.011
Asthma Duration ^1^	0.125	0.44	0.061
ICS daily dose ^1^	−0.087	0.17	0.366
SABA daily dose ^1^	−0.022	0.09	0.554
FEV_1_% ^1^	−0.246	0.77	0.028
Correlation of asthma and left ventricular hypertrophy			
Asthma ^1^	0.131	0.32	0.112
Severe Asthma ^1^	0.136	0.38	0.095
Asthma Duration ^1^	0.090	0.19	0.288
ICS daily dose ^1^	0.071	0.17	0.341
SABA daily dose ^1^	0.048	0.11	0.488
FEV_1_% ^1^	−0.198	0.55	0.055

^1^ Multivariate analysis adjusted for: essential hypertension, hypertension severity, diabetes, Body Mass Index, and creatinine clearance. ICS: Inhaled corticosteroid; SABA: Short-acting beta-agonist; FEV_1_: Forced expiratory volume in the 1st second.

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
