# Peer review of "Echocardiographic Evaluation of the Cardiac Chambers in Asthmatic Patients: The BADA (Blood Pressure Levels, Clinical Features and Markers of Subclinical Cardiovascular Damage of Asthma Patients) Study-ECO"

_jpm, 2022, doi:10.3390/jpm12111847_

Round 1

Reviewer 1 Report

The authors have addressed a very important issue regarding cardiac changes in asthmatic patients. The manuscript is well presented. However, there are few minor concerns as follows

1. What was the reason to choose one 1-std Dev instead of 2.

2. Is there any specific reasons to enroll only Caucasian population.

3. As mentioned elsewhere, DM, HTN, and increased BMI itself exert individual effects on cardiac functions, why these population was not excluded from the study to minimize the bias.

4. A figure having main echo findings may be included for the interest of readership. 

Author Response

REVIEWER #1

We would like to thank you for the review of our manuscript and for your positive opinion about it. We greatly appreciate the effort in revising our manuscript; we have accepted all your suggestions and revised the article according to them.

The authors have addressed a very important issue regarding cardiac changes in asthmatic patients. The manuscript is well presented. However, there are few minor concerns as follows:

What was the reason to choose one 1-std Dev instead of 2.

Thank you for the clarification: for a population such as the one enrolled in our study, the submission of the results as mean +/- 1 standard deviation is generally adequate to represent the statistical variability of the sample.

Is there any specific reasons to enroll only Caucasian population.

Caucasian ethnicity is not actually an inclusion criterion in our study. We decided to specify the ethnicity of the enrolled population to avoid any confounding factor given the presence of small percentages of non-Caucasian patients attending our outpatient facilities

As mentioned elsewhere, DM, HTN, and increased BMI itself exert individual effects on cardiac functions, why these population was not excluded from the study to minimize the bias.

Thank you for raising this important point. The objective of the BADA Study is, among others, to analyze the level of cardiovascular involvement (subclinical and clinical) of asthmatic versus non-asthmatic patients and to quantify non-pulmonary organ damage. Our data seem to indicate a cardiac involvement that, as shown by logistic regression analysis, is associated with asthma disease itself rather than with the existence of any other known cardiovascular risk factors. In this sense, the evaluation of patients who are also affected by cardiovascular risk factors does not represent a true bias, since the analyses performed on our data allow us to suggest that the presence of hypertension, diabetes mellitus, or elevated BMI would act in the asthmatic as a concomitant in determining cardiac damage rather than as a primary cause, as might be assumed.

We have changed the discussion section (lines 369-373) according to your suggestion.

A figure having main echo findings may be included for the interest of readership.

Thank you for the suggestion. In the revised version of our manuscript we added a figure displaying the list and the main features of the echo findings evaluated in the present study

We hope that we have successfully changed our manuscript according to your suggestions and that we have provided all the necessary explanations. We also hope that the manuscript now fulfills your criteria, and the Journal criteria for publication.

Reviewer 2 Report

This research revealed characteristic changes of UCG parameters in asthma patients.

As the authors pointed out, the asthma patients have higher blood pressure, higher BMI, higher prevalence of DM, and lower creatinine clearance. It is well known that hypertension and DM cause diastolic dysfunction, therefore the changes in UCG parameters may be just reflecting hypertension or DM which patients had unrelated to asthma.

Line 305: "increased LV mass appears to be one of the pivotal elements of the cardiac remodeling in asthma patients." It could be true, however, asthma patients in this study showed rather high BMI (28.11+-4.09).  Isn't higher BMI related to increased LV mass, besides asthma ?

Another issue is dose of SABA and ICS. It is known that excessive use of SABA deteriorates cardiac function. The authors should describe the dose of these inhalers in the study.

Author Response

REVIEWER #2:

We would like to thank you for your careful review of our manuscript. We greatly appreciate your evaluation and the helpful suggestions provided that will certainly improve its quality. We put a lot of effort in this study, and we appreciate your opinion very much.

This research revealed characteristic changes of UCG parameters in asthma patients.

As the authors pointed out, the asthma patients have higher blood pressure, higher BMI, higher prevalence of DM, and lower creatinine clearance. It is well known that hypertension and DM cause diastolic dysfunction, therefore the changes in UCG parameters may be just reflecting hypertension or DM which patients had unrelated to asthma.

Thank you for raising this important point. The objective of the BADA Study is, among others, to analyze the level of cardiovascular involvement (subclinical and clinical) of asthmatic versus non-asthmatic patients and to quantify non-pulmonary organ damage. Our data seem to indicate a cardiac involvement that, as shown by logistic regression analysis, is associated with asthma disease itself rather than with the existence of any other known cardiovascular risk factors. In this sense, the evaluation of patients who are also affected by cardiovascular risk factors does not represent a true bias, since the analyses performed on our data allow us to suggest that the presence of hypertension, diabetes mellitus, or elevated BMI would act in the asthmatic as a concomitant in determining cardiac damage rather than as a primary cause, as might be assumed.

We have changed the discussion section (lines 369-373) according to your suggestion.

Line 305: "increased LV mass appears to be one of the pivotal elements of the cardiac remodeling in asthma patients." It could be true, however, asthma patients in this study showed rather high BMI (28.11+-4.09). Isn't higher BMI related to increased LV mass, besides asthma ?

We want to thank the reviewer for raising this important concern. In the revised version we have more extensively discussed how much the increased BMI in asthmatic patients may have actually affected the alterations in cardiac structure and function that we showed in our study (lines 355-366).

Another issue is dose of SABA and ICS. It is known that excessive use of SABA deteriorates cardiac function. The authors should describe the dose of these inhalers in the study.

Thank you for this important comment. This is another critical issue. In our previous study, we did not show a direct relationship between chronic intake of inhaled drugs (both LABA and ICS) and the development of arterial hypertension even when comparing high dosages with low dosages [Di Raimondo D, et al. Int. J. Environ. Res. Public Health 2020, 17, 6925. https://doi.org/10.3390/ijerph17186925]. Likewise, other similar studies have yielded inconclusive results. For example, Wang MT, et al. in a large case-control study of more than 280.000 patients with COPD [Wang MT, Liou JT, Lin CW, Tsai CL, Wang YH, Hsu YJ, Lai JH. Association of Cardiovascular Risk With Inhaled Long-Acting Bronchodilators in Patients With Chronic Obstructive Pulmonary Disease: A Nested Case-Control Study. JAMA Intern Med. 2018 Feb 1;178(2):229-238. doi: 10.1001/jamainternmed.2017.7720.], conclude that “overall use of inhaled long-acting bronchodilators was not associated with an increased risk of CVD across different recency of therapy, although a 10% decrease in cardiovascular risk was observed with past LABA use”. The mean LABA daily dose (mcg) has been added in table 1. The discussion section has been improved deepening this issue.

We hope that we have successfully changed our manuscript according to your suggestions and that we have provided all the necessary explanations. We also hope that the manuscript now fulfills your criteria, and the Journal criteria for publication.

Reviewer 3 Report

This is a clinical study regarding cardiac dysfunction in asthma patients. The authors concluded that a more in-depth cardiovascular evaluation in patients with chronic respiratory disease, particularly in asthmatic patients, allows the identification of a significant subclinical level of cardiac involvement.

This reviewer considers that this is clinically important. Asthma can be included as one of cardiac risk factors, and asthma patients are in stage A of heart failure. However, this reviewer considers that much more information are required to examine this issue. 

Major comments:

1.     In the development of cardiac dysfunction, life-style diseases are very important. Thus, the data regarding blood pressure and heart rate should be included.

2.     In Table 1, more hypertension patients were enrolled in asthma patients compared with control, and asthma and severe asthma remained significant after HT adjusted, as shown in Table 3a and 3b. This reviewer considers that it may not enough. Hypertension severity should be adjusted. 

3.     If the authors examine if the asthma itself affects cardiac function, they should exclude hypertension patients. 

4.     Also, anemia is important in cardiac function. Red blood cell data should be included in Table 1. 

5.     Lung function data, other than FEV1 should be included in Table 1.

6.     Body mass index, mostly meaning obesity, also affects cardiac function. The authors also should adjust BMI. 

7.     Lines 233-234. The authors described that hypertension was not significantly different between the 2 groups, but p-value was 0.01. 

8.     The authors should show the demonstrable pictures of echocardiography in both 2 groups. 

9.     How do the authors consider about hidden factors, which may affect cardiac function?

Author Response

REVIEWER #3:

We would like to thank you for your careful review of our manuscript. We greatly appreciate your evaluation and the helpful suggestions provided that will certainly improve its quality. We put a lot of effort in this study, and we appreciate your opinion very much.

This is a clinical study regarding cardiac dysfunction in asthma patients. The authors concluded that a more in-depth cardiovascular evaluation in patients with chronic respiratory disease, particularly in asthmatic patients, allows the identification of a significant subclinical level of cardiac involvement.

This reviewer considers that this is clinically important. Asthma can be included as one of cardiac risk factors, and asthma patients are in stage A of heart failure. However, this reviewer considers that much more information are required to examine this issue.

Major comments:

  1. In the development of cardiac dysfunction, life-style diseases are very important. Thus, the data regarding blood pressure and heart rate should be included.

Thank you for the important remark. We added data on systolic blood pressure, diastolic blood pressure, and heart rate in Table 1. The statement: “All subjects enrolled were self-reported as being sedentary” has been added (lines 100-101).

  1. In Table 1, more hypertension patients were enrolled in asthma patients compared with control, and asthma and severe asthma remained significant after HT adjusted, as shown in Table 3a and 3b. This reviewer considers that it may not enough. Hypertension severity should be adjusted.

We want to thank the reviewer for raising this important concern. Taking into consideration your suggestion, we repeated our multiple logistic regression analysis by adding among the possible confounders, as an indicator of "hypertension severity," the “uncontrolled hypertension” i.e., the failure to achieve the targets recommended by the most recent ESC-ESH guidelines (Williams B, Mancia G, Spiering W et al. 2018 ESC/ESH Guidelines for the management of arterial hypertension. Eur Heart J. https://doi.org/10.1093/eurheartj/ehy339). Adjusting the multiple logistic regression analysis concordantly, the results (presented in the revised version of our manuscript) are almost unchanged, confirming that high blood pressure levels, taking also into account  hypertension severity, seem not be the major determinant of cardiac involvement in our sample.

  1. If the authors examine if the asthma itself affects cardiac function, they should exclude hypertension patients.

Thank you for raising this important point. The objective of the BADA Study is, among others, to analyze the level of cardiovascular involvement (subclinical and clinical) of asthmatic versus non-asthmatic patients and to quantify non-pulmonary organ damage. Our data seem to indicate a cardiac involvement that, as shown by logistic regression analysis, is associated with asthma disease itself rather than with the existence of any other known cardiovascular risk factors. In this sense, the evaluation of patients who are also affected by cardiovascular risk factors does not represent a true bias, since the analyses performed on our data allow us to suggest that the presence of hypertension, diabetes mellitus, or elevated BMI would act in the asthmatic as a concomitant in determining cardiac damage rather than as a primary cause, as might be assumed.

We have changed the discussion section (lines 369-373) according to your suggestion.

  1. Also, anemia is important in cardiac function. Red blood cell data should be included in Table 1.

Thank you for the suggestion. In the revised version of our manuscript we added data on Red Blood Cells count, Hemoglobin and Mean corpuscular volume (MCV) levels.

  1. Lung function data, other than FEV1 should be included in Table 1.

Thank you for the suggestion. In the revised version of our manuscript we added data on the other two key spirometry measures that allow us to quantify the level of obstruction in asthmatic subjects: Forced Vital Capacity (FVC) and FEV1/FVC ratio.

  1. Body mass index, mostly meaning obesity, also affects cardiac function. The authors also should adjust BMI.

We agree with the reviewer. The logistic regression analysis presents data that were also adjusted for BMI values.

  1. Lines 233-234. The authors described that hypertension was not significantly different between the 2 groups, but p-value was 0.01.

Thank you comment. The results section, analyzing table 1, states that “With a comparable family history of hypertension, the percentage of asthmatics with hypertension was significantly higher than controls: 57.5% (50 subjects) vs 35.0% (35 subjects) (p:0.01). The two groups did not differ in relation to the main comorbidities/risk factors associated with hypertension, except for a higher prevalence of type 2 Diabetes Mellitus (DM) in the asthma group (8 patients vs 0) (p:0.001), despite a concordance of fasting blood glucose values in asthmatics vs controls (90. 9 vs 86.9 mg/dL) (p:0.07)”.

  1. The authors should show the demonstrable pictures of echocardiography in both 2 groups.

We thank the reviewer for the suggestion. The transthoracic echocardiogram is a dynamic examination, and many of the measurements made would not be clearly visible in the screenshots taken during an examination that lasts at least half an hour and is subsequently processed by the operators to extrapolate many of the indices we present in our study. In accordance also with the suggestions of another reviewer, we have added a figure to the revised version of the manuscript that better explains the echocardiographic assessments presented. We understand that the reviewer's indication was in the reader's interest in order to make the interpretation of the results clearer, but we hope that the revised version of our manuscript will be sufficiently improved in this way as well

  1. How do the authors consider about hidden factors, which may affect cardiac function?

Thank you for raising this point. It is of course possible that other factors not directly addressed in our analysis may have influenced the results, although our investigation was quite thorough. We have added this possibility among the limitations of our study

We hope that we have successfully changed our manuscript according to your suggestions and that we have provided all the necessary explanations. We also hope that the manuscript now fulfills your criteria, and the Journal criteria for publication.

Round 2

Reviewer 2 Report

The authors replied to my comments and revised accordingly.

However, as the authors admitted, this research has multiple major limitations.  They failed to match the study group (asthma group) and control group regarding major confounding factors such as BMI, hypertension, and diabetes.  Another major limitation is the use of SABA which could have deteriorated cardiac function depending on the doses. In the Line 410-412 the authors wrote "Our study seems to indicate that both ICS and SABA inhaled therapy does not affect the possible myocardial involvement." The P values for SABA in Table 3 are nearly 0.5.  The authors should address more about these confounding factors.

The abstract must be revised to avoid misleading.

Author Response

REVIEWER #2:

We would like to thank you for your further evaluation of our manuscript. We greatly appreciate your helpful suggestions provided that will certainly improve its quality.

The authors replied to my comments and revised accordingly.

However, as the authors admitted, this research has multiple major limitations.  They failed to match the study group (asthma group) and control group regarding major confounding factors such as BMI, hypertension, and diabetes.  Another major limitation is the use of SABA which could have deteriorated cardiac function depending on the doses. In the Line 410-412 the authors wrote "Our study seems to indicate that both ICS and SABA inhaled therapy does not affect the possible myocardial involvement." The P values for SABA in Table 3 are nearly 0.5.  The authors should address more about these confounding factors.

The abstract must be revised to avoid misleading.

Thank you for these important comments.

About this we would like to add that the two groups examined included patients without recent history of cardiovascular events or history of conditions capable of altering cardiac structure or function. In addition, the excess cardiovascular risk observed in the asthma group is limited to hypertension (whose role has been thoroughly analyzed and will be further studied in the BADA Study) and overweight. Both groups had overlapping age, overlapping lipid profile, and did not enroll active smokers. Regarding diabetes, although in the asthma group the % of diabetics is significantly higher than the control group this (about 9 %) is superimposable to the general adult population in Italy in 2021 (https://www.salute.gov.it). These elements in our view justify, at least partially, our conclusions, which certainly deserve to be verified and validated in larger studies.

Regarding the role of SABAs, although these have been the subject of numerous analyses and are still under special observation, many studies, including ref #22 of our paper but also Nwaru BI, et al. (Overuse of short-acting β2-agonists in asthma is associated with increased risk of exacerbation and mortality: a nationwide cohort study of the global SABINA program. Eur Respir J. 2020 Apr 16;55(4):1901872), for example, acknowledging an additional risk for overuse of SABAs (which in any case was not present in our sample), do not attribute it to excess of cardiovascular risk due to structural heart damage. Caution in the use of these drugs is, however, mandatory.

In the second revision of our manuscript, we tried to emphasize even more the possible limitations of our study, as suggested by the reviewer, by revising the abstract and discussion section. We fully agree with the reviewer that the limitations of our study need to be more clearly mentioned, and we have amended the manuscript accordingly commenting our results more objectively.

We hope that we have successfully changed our manuscript according to your suggestions and that we have provided all the necessary explanations. We also hope that the manuscript now fulfills your criteria, and the Journal criteria for publication.

Reviewer 3 Report

This reviewer considers that this paper was well revised. This reviewer has no further comment. 

Author Response

We again thank the reviewer for significantly helping to improve the quality of our manuscript with his comments and suggestions